# Therapeutic Interaction of Apatinib and Chidamide in T-cell Acute Lymphoblastic Leukemia through Interference with Mitochondria Associated Biogenesis and Intrinsic Apoptosis

**DOI:** 10.3390/jpm11100977

**Published:** 2021-09-29

**Authors:** Mengya Zhong, Fusheng Lin, Yuelong Jiang, Guangchao Pan, Jinshui Tan, Hui Zhou, Qian Lai, Qinwei Chen, Manman Deng, Jie Zha, Bing Xu

**Affiliations:** 1Department of Hematology, The First Affiliated Hospital of Xiamen University and Institute of Hematology, School of Medicine, Xiamen University, Xiamen 361003, China; doczmyya@stu.xmu.edu.cn (M.Z.); ailefu@stu.xmu.edu.cn (F.L.); jingly@stu.xmu.edu.cn (Y.J.); panguangchao@stu.xmu.edu.cn (G.P.); 24520201154265@stu.xmu.edu.cn (J.T.); 21620171152459@stu.xmu.edu.cn (H.Z.); mlcqlai@stu.xmu.edu.cn (Q.L.); chenqinwei1212@stu.xmu.edu.cn (Q.C.); marina_deng@outlook.com (M.D.); xmdyJay@outlook.com (J.Z.); 2Key Laboratory of Xiamen for Diagnosis and Treatment of Hematological Malignancy, No. 55, Shizhen Hai Road, Xiamen 361003, China

**Keywords:** T-cell acute lymphoblastic leukemia (T-ALL), apatinib, chidamide, synergy, mitochondria, citric acid cycle or tricarboxylic acid cycle (TCA), oxidative phosphorylation (OXPHOS)

## Abstract

T-cell acute lymphoblastic leukemia (T-ALL) shows poor clinical outcome and has limited therapeutic options, indicating that new treatment approaches for this disease are urgently required. Our previous study demonstrated that apatinib, an orally selective VEGFR-2 antagonist, is highly effective in T-ALL. Additionally, chidamide, a histone deacetylase inhibitor, has proven to be cytotoxic against T-ALL in preclinical and clinical settings. However, whether the therapeutic interaction of apatinib and chidamide in T-ALL remains unknown. In this study, apatinib and chidamide acted additively to decrease cell viability and induce apoptosis in T-ALL in vitro. Notably, compared with apatinib or chidamide alone, the combinational regimen was more efficient in abrogating the leukemia burden in the spleen and bone marrow of T-ALL patient-derived xenograft (PDX) models. Mechanistically, the additive antileukemia effect of apatinib and chidamide was associated with suppression of mitochondrial respiration and downregulation of the abundance levels of several rate-limiting enzymes that are involved in the citric acid cycle and oxidative phosphorylation (OXPHOS). In addition, apatinib enhanced the antileukemia effect of chidamide on T-ALL via activation of the mitochondria-mediated apoptosis pathway and impediment of mitochondrial biogenesis. Taken together, the study provides a potential role for apatinib in combination with chidamide in the management of T-ALL and warrants further clinical evaluations of this combination in patients with T-ALL.

## 1. Introduction

T-cell acute lymphoblastic leukemia (T-ALL) is an aggressive hematological tumor arising from malignant transformation and clonal expansion of T cell progenitors [1]. Patients with T-ALL represent almost 15% and 25% of pediatric and adult ALL cases and show dismal clinical outcomes [2,3,4]. Treatment failure mainly attributes to either primary or acquired chemoresistance, while there are limited therapeutic approaches for the management of chemoresistant T-ALL patients. Therefore, these observations suggest that developing novel therapeutic strategies to combat this life-threatening disease is an unmet medical need.

Angiogenesis is an important component involved in the tumorigenesis of various neoplasms, encompassing ALL [5,6,7]. Multiple proangiogenic mediators are implicated in the process of angiogenesis, among which vascular endothelial cell growth factors (VEGFs) and their cognate receptors (VEGFRs) play vital roles in this process. The targeting VEGF/VEGFR pathway, therefore, has become prevalent as antitumoral regimen and has shown moderate cytotoxic effects on diverse tumor types, including leukemia [8,9]. Apatinib, a receptor tyrosine kinase inhibitor, selectively targets VEGFR2 and has shown antitumor efficacy with manageable side effects [10,11,12]. It has been approved as a safe and effective drug for patients with advanced gastric cancer who have failed to frontline chemotherapy. Moreover, data uncovered that apatinib has exhibited antitumor effects on multiple solid tumors of ongoing phase II and III clinical trials [12,13]. In addition, apatinib could sensitize tumor cells to the cytotoxicity of other anticancer drugs, suggesting that this drug might be a potential chemotherapy sensitizer [14,15]. With respect to its activity against T-ALL, our prior study verified that apatinib alone was effective in inhibiting cell viability and trigger apoptosis in T-ALL cellular models [16]. In general, prescribing a particular drug alone is insufficient to completely eradicate T-ALL cells and achieve durable remission, thus combining apatinib with other compounds is necessary for elimination of T-ALL cells. 

Apart from genetic lesions, epigenetic alterations are also identified as important contributors for the development of ALL [17,18]. Histone acetylation aberration is a frequent epigenetic event that modulates by the enzymes with histone acetyltransferase (HAT) or histone deacetylase (HDAC) activity [19,20]. Malignant diseases, including ALL, upregulate the expression of HDACs and enhance its activity, making HDACs attractive drug targets. Of importance, histone deacetylation inhibition with HDAC inhibitors (HDACi) has been proven to be a promising antitumor approach for the treatment of multiple tumor types, especially for peripheral T-cell lymphomas (PTCL) [21,22]. Unlike PTCL, T-ALL shows less vulnerable to the cytotoxicity of HDAC inhibitors [23,24,25], suggesting that combination with other compounds is necessary to potentiate the efficacy of HDACi against T-ALL. Chidamide, also known CS055 or HBI-8000, is a new blocker of HDAC1, 2, 3, and 10 and shows encouraging therapeutic responses in patients with relapsed and refractory PTCL. Based on its clinical therapeutic responses in PTCL, chidamide has received an approval for the treatment of this special disease population in China [26,27,28,29]. Chidamide monotherapy or chidamide-based regimens exert moderate or significant antileukemia effects on T-ALL preclinical and clinical models [16,25,30,31]. However, the therapeutic effect of apatinib combined with chidamide on T-ALL has not yet been investigated.

In this study, we revealed that apatinib enhanced the antileukemia efficacy of chidamide against T-ALL cells in vitro and *in vivo*. The additive effect of both drugs was associated with blockade of mitochondrial respiration and downregulation of rate-limiting enzymes involved in the citric acid cycle and oxidative phosphorylation (OXPHOS). Additionally, apatinib and chidamide was additive to kill T-ALL via activation of the mitochondrial apoptosis pathway.

## 2. Materials and Methods

### 2.1. Reagents

Apatinib was kindly provided by Jiangsu Hengrui Medicine Company (Lianyungang, Jiangsu, China). Chidamide (purity > 95%) was supplied by Chipscreen Bioscience Ltd. (Shenzhen, China). Both agents were dissolved in dimethyl sulfoxide (DMSO, Invitrogen, Carlsbad, CA, USA) to obtain a stock solution of 100 mM. All the above stock solutions were stored at −20 °C. In all in vitro assays, the two drugs were diluted into designated concentrations with culture medium.

### 2.2. Cell Culture

Human T-ALL cell lines, Molt-4 and Jurkat cells, were obtained from the Department of Hematology, Key Laboratory of Xiamen for Diagnosis and Treatment of Hematological Malignancy (Xiamen, Fujian, China). Both Molt-4 and Jurkat cells were routinely cultured in RPMI 1640 medium (HyClone, Logan, UT, USA) supplemented with 10% fetal bovine serum (FBS, Gibco, Grand Island, NY, USA) and 1% penicillin/streptomycin (Invitrogen) at 37 °C incubator with 5% CO_2_.

### 2.3. Cell Viability Assay

Cell viability of Molt4 and Jurkat cells exposed to chidamide with or without apatinib were determined by Cell Counting Kit-8 (CCK-8, MedChemExpress, Shanghai, China) assay. In brief, T-ALL cells (5 × 10^4^ cells per well) were seeded in 96-well plates and treated with designated doses of chidamide alone or in combination with 5 μM apatinib for 48 h and 72 h, then added CCK-8 reagents (10 μL per well) and continued to incubate for additional 2 h. At the end of the assay, the absorbances were detected at 450 nm by iMark Microplate Absorbance Reader (Bio-rad, Hercules, CA, USA). The IC50 value (half maximal inhibitory concentration) of each cell line was calculated using the GraphPad Prism 8 software (San Diego, CA, USA). All experiments were repeated at least three times with triplicate in each test.

### 2.4. Flow Cytometry Analyses

To assess apoptosis, Molt4 and Jurkat cells were cultured with chidamide alone or in combination with 5 μM apatinib for 48 h and 72 h, then double labeled with Annexin-V-FITC/PI staining (eBioscience, San Diego, CA, USA) for 15 min at room temperature in the dark according to the manufacturer’s instructions. The stained cells were analyzed by flow cytometry (Beckman Coulter, Brea, CA, USA). Apoptotic cells were defined as Annexin V positive cells.

To evaluate conversion of transform in reactive oxygen species (ROS) for cells, changes in cellular ROS levels were determined using a Reactive Oxygen Species Assay Kit (Beyotime, Shanghai, China) as previously shown. For each sample, the basal values were subtracted from those obtained after different treatments and results were presented as the ratio of the mean fluorescence intensity. We conducted three independent experiments to determine the apoptosis and ROS levels, and the data indicated as mean ± SD.

### 2.5. Metabolic and Biochemical Assays

Intracellular ATP content was quantified with the enhanced ATP assay kit according to the user manual (Beyotime). After cells (5 × 10^5^ per test) were lysed, ATP concentration in the supernatant was quantified by measuring the fluorescence emitted by firefly luciferase on a luminometer (Berthold, Germany). The concentration of intracellular ATP was shown in nmol/mg protein. Total protein was quantified for normalization using the Bio-Rad Protein Assay (Bio-Rad).

Citrate concentration was determined with a citrate assay kit following the manual (Abcam, Cambridge, UK). Briefly, the assay buffer was used to collect cells (2 × 10^6^ per test), and supernatants were incubated with 50 μL citrate reaction mixture containing an enzyme mix, the developer and the citrate probe for 30 min at room temperature. Absorbance was measured at 570 nm on an Absorbance Reader (Bio-Rad). Citrate concentration in the supernatant was extrapolated from a standard curve and normalized to cell number. Each assay was performed in at least three technical replicates of independent experiments.

### 2.6. Oxygen Consumption Assay 

According to the manufacturer’s instructions, the cellular oxygen consumption rate (OCR) was measured on a Seahorse XF Extracellular Flux Analyzer (Seahorse Bioscience, Santa Clara, MA, USA). After treatment with 3 µM chidamide and/or 5 µM apatinib for 24 h, Molt4 and Jurkat cells (3 × 10^5^ cells per well) were resuspended in XF media and plated in XFe-96 plates in XF media. Real-time measurement of OCR was performed on an XFe-96 Extracellular Flux Analyzer (Seahorse Bioscience). OCR was measured in XF medium (non-buffered DMEM medium containing 10 mM glucose and 1 mM sodium pyruvate) under basal conditions, as well as in response to 1 μM oligomycin, 1 μM of FCCP (carbonylcyanide-4-(trifluoromethoxy)-phenylhydrazone) and 1 μM of antimycin and rotenone (Sigma-Aldrich, St. Louis, MO, USA), respectively. The experiment using the Seahorse system was performed under the following conditions: mixture, 3 min; wait, 3 min; measurement time, 3 min.

### 2.7. Western Blot Analysis

Molt4 and Jurkat cells were cultured for 48 h in the absence or presence of 5 μM apatinib and 0.75 μM chidamide, and harvested and resuspended in RIPA lysis buffer containing 1% protease inhibitor mixture and 1% phenylmethanesulfonyl fluoride (APExBIO, Houston, TX, USA). Total protein (20–30 μg) were separated by 8–12% gel electrophoresis and transferred PVDF membrane (Millipore, Burlington, MA, USA). The protein expression levels were probed with primary antibodies and HRP-conjugated secondary antibodies (1:2000, Cell Signaling, Herts, UK). The primary antibodies including Bad (CA9292S), Bcl2 (CA15071S), Bcl-xL (CA2762S), XIAP (CA2042S), cytochrome C (CA4272S) were diluted at 1:2000 in TBST. Anti-GAPDH antibody (1:5000, Proteintech, Suite, USA) was used as loading control. Proteins were visualized using using a hypersensitive ECL chemiluminescence kit (NCM Biotech, Suzhou, China).

### 2.8. RNA Isolation and Real-Time Quantitative PCR

Total cellular RNA was isolated using a SteadyPure Universal RNA Extraction Kit (AG21017, Accurate Biology, Changsha, Hunan, China) following the manufacturer’s protocol. An Evo M-MLV RT Premix for qPCR Kit (AG11706, Accurate Biology) was used to generate cDNA from 1 µg of total RNA. Quantitative real-time PCR (qRT-PCR) was performed with SYBR Green Premix Pro Taq HS qPCR Kit II (AG11702, Accurate Biology), and detection used a Light Cycler 480 System (Roche, Basel, Switzerland). The qRT-PCR reactions were performed at 95 °C for 30 s, followed by 40 cycles of 95 °C for 5 s and 60 °C for 30 s. The melting curve was generated from 60 °C to 95 °C at a 0.3 °C increase per 15 s. The ΔΔCt method was used for analysis, with actin as an endogenous reference. All the primers used in the study are listed in Appendix A, and experiments were repeated three times.

### 2.9. In Vivo Tumor Growth Assay

All animal experiments were performed in accordance with the guidelines of the Animal Welfare Committee of Xiamen University (Fujian, China). NOD-^Prkdc−/−^IL2^rg−/−^ mice (NSG, 4–6 weeks old, male) were purchased from Beijing NPI IDMO ltd. and housed under specific pathogen-free (SPF) condition in Xiamen University Laboratory Animal Center. Primary T-ALL cells were collected from a T-ALL patient with informed consent and were used to establish patient-derived xenograft (PDX) mouse model. The clinical characteristics of this patient are presented in Appendix A. 

The study protocol was approved by the Ethics Committee of Xiamen University complying with the Declaration of Helsinki and stated below. Briefly, NSG mice received 1 Gy of irradiation and were subsequently implanted with primary T-ALL cells (2 × 10^6^ cells/mouse) intravenously which were obtained from the established first-generation T-ALL PDX model mouse. Human CD45 (hCD45) staining (clone HI30, Biolegend, CA, USA) in peripheral blood (PB), a marker of leukemia burden, was tested using flow cytometry once weekly. Once the percentage of hCD45 in PB ≥ 1%, these PDX mice were randomized to the following four groups (n = 3 per group) for a successive 14-day treatment schedule: (1) control group (normal saline); (2) apatinib group (100 mg/kg/day); (3) chidamide group (10 mg/kg/day); (4) apatinib plus chidamide group. At the end of the experiment, all PDX mice were euthanized and their bone marrow (BM) and spleen (SP) samples were separately harvested and used to analyze leukemia burden using flow cytometry. Leukemia burden in BM and SP was defined as human CD45-positive and mouse CD45-negative cells. hCD45 cells was detected with anti-human CD45 antibody (clone HI30, Biolegend, CA, USA) and mouse CD45 with BV421 Rat Anti-Mouse CD45 antibody (563890, BD Biosciences, Franklin Lakes, NJ, USA). 

### 2.10. Histological Analysis

Liver samples from xenografted mice were formalin-fixed, and paraffin-embedded and sections were subjected to xylene dewaxing, a portion was used for hematoxylin and eosin (HE) staining, and the remaining sections were immunostained according to a manual method using custom-made and validated antibodies as follows: CD45 (1:200, CA13917, Cell Signaling), Ki67 (1:200, ab15580, Abcam) and PCNA (1:200, ab18197, Abcam). DAB (DAB-2032, MXB Biotechnologies, Fujian, China) was applied for 5 min at room temperature according to the manufacturer’s protocols. The staining pattern was semiquantitatively assessed based on staining distribution, and the results were correlated with morphological variables. Subsequently, the remaining sections were used for immunofluorescence staining, with TUNEL-FITC (A111-03, Vazyme Biotech, Nanjing, Jiangsu, China) applied at room temperature for 20 min. Analysis was performed under a fluorescence microscope (Nikon, Eclipse Ci-L, Japan). Staining signals were scored based on the percentage of positive tumor cells. Image-Pro Plus 6.0 (Media Cybernetics, Inc., Bethesda, MD, USA) was used to analyze the positive cell percentages of the immunohistochemistry and immunofluorescence staining. Positive rate (%) = number of positive cells/number of total cells * 100.

### 2.11. Statistical Analysis

All data in this research were expressed as mean ± standard deviation (SD) from three independent experiments. Line charts or the corresponding bar graphs were drawn with the GraphPad Prism 8 software (USA). An unpaired Student’s t-test was used for group pair comparisons. A one-way ANOVA followed by Bonferroni post-hoc test was conducted for multiple groups. *p* < 0.05 was considered statistically significant.

## 3. Results

### 3.1. The Additive Antileukemia Activity of Apatinib in Combination with Chidamide in T-ALL Cells

In this study, we sought to investigate whether apatinib might impact the antitumoral efficacy of chidamide against T-ALL in preclinical models. Jurkat and Molt4, two T-ALL cell lines, were seleceted and cultured with various concentrations of chidamide with or without apatinib (5 μM) for 24 h, 48 h and 72 h. Cell viability was then assessed with a CCK8 kit. For 24 h treatment, chidamide with or without apatinib only resulted in slight effects on loss of cell viability in both cell lines (Appendix A). Inconsistently, exposing T-ALL cells to chidamide alone led to dose- and time-dependent cell viability decrease, and this effect was further enhanced when combining chidamide with apatinib for 48 h and 72 h treatment (Figure 1A,B). The IC50 values were shown in Table 1. Chidamide combined with apatinib showed significantly lower IC50 values in comparison with chidamide alone in Jurkat cells, regardless of treatment duration (Table 1). Analogous results were observed in Molt4 cells treated with chidamide alone or in combination with apatinib for 48 and 72 h.

To further interrogate the cell-killing activity of chidamide plus apatinib, an Annexin V/PI kit was introduced to examine the apoptotic cells of Molt4 and Jurkat treated as described in cell viability assay. As expected, chidamide alone induced dose- and time-dependent apoptosis enhancement in the two T-ALL cells (Figure 2 and Appendix A). Consistent with the cell viability results, chidamide in the presence of apatinib further enhanced apoptotic cell deaths in each cell line compared with chidamide alone. Table 2 and Table 3 showed the apoptotic percentages of Jurkat and Molt4 cells exposed to chidamide with or without apatinib for indicated timepoints. Taken together, these data unravel that apatinib and chidamide are additive to blunt cell viability and promote apoptosis in T-ALL cells.

### 3.2. The Cytotoxic Effects of Apatinib and Chidamide on T-ALL Cells Are Associated with Blockade of Mitochondrial Respiration 

Metabolic pathways are often reprogrammed to support cell growth and survival in malignant tumors [32]. Evidence has been validated that mitochondrial oxidative phosphorylation (OXPHOS), rather than aerobic glycolysis, might be the main source of ATP production in T-ALL [33]. To demonstrate whether mitochondrial function is affected by apatinib combined with chidamide, oxygen consumption rate (OCR) is measured on a Seahorse extracellular flux analyzer. Both apatinib-treated and chidamide-treated cells showed lower mitochondrial respiration compared with untreated T-ALL cells, as evidenced by reduced basal and maximal OCR levels. Such effects on basal and maximal OCR levels were more robust in the combined group than in the other three groups, indicative of lower oxidative metabolism (Figure 3A,B). Accordingly, each single drug had lower ATP production than the untreated group, while cotreatment of T-ALL cells with apatinib coupled with chidamide exhibited the lowest levels of ATP production (Figure 3C,D). These results indicate that mitochondrial respiration suppression is implicated in the additive antitumor efficacy of apatinib coupled with chidamide on T-ALL.

### 3.3. Cotreatment of Apatinib and Chidamide Interferes with the Citric Acid Cycle

Citric acid cycle is the well-known tricarboxylic acid cycle (TCA) critical for mitochondrial aerobic respiration [34,35,36]. Whether respiration reduction caused by the combined regimen influences the TCA cycle or not remains unknown. Compared with no drug treatment, either chidamide or apatinib resulted in lower citric acid levels, the first product of TCA, in both Jurkat and Molt4 cells. Of importance, the combined treatment further diminished citric acid production compared with the other three groups in both T-ALL cell lines (Figure 4A,B). 

TCA is under the control of a series of rate-limiting enzymes, including PDHA1, MPC1, CS, SDHD, MDH1 and COX10. We carried out qRT-PCR assays to study the influence of apatinib and chidamide alone or in combination on the mRNA expression levels of these enzymes in T-ALL cells. Cotreatment of T-ALL cells with apatinib and chidamide significantly downregulated the mRNA levels of these enzymes, whereas each single drug had subtle to moderate effects on downregulation of these tested enzymes (Figure 4C,D). Collectively, these findings reveal that perturbation of the process of TCA contributes to the antileukemia interaction of apatinib and chidamide against T-ALL cells.

### 3.4. Apatinib Potentiates Chidamide Induced Apoptosis via Deregulation of Anti- and Pro-Apoptotic BCL-2 Family Components

Reactive oxygen species (ROS) are the main byproduct of mitochondrial respiration [37,38]. To address whether the drug combination could affect ROS production, intracellular ROS levels were analyzed in Jurkat and Molt4 cells exposed to chidamide in the presence or absence of apatinib. Treatment with chidamide or apatinib alone had no impact on the production of ROS, while their combination regimen moderately decreased ROS generation (Figure 5A,B). This observation was in line with the findings of blocking mitochondrial respiration in the chidamide plus apatinib treatment. 

Antiapoptotic and proapoptotic BCL2 family proteins are the predominant regulators of mitochondrial apoptosis, so-called intrinsic apoptosis [39]. Due to the function of apatinib combined with chidamide in mitochondria associated processes, we speculated that this combination might interfere with the expression of BCL2 family components. As anticipated, apatinib and chidamide cooperated to downregulate the expression of several antiapoptotic BCL2 proteins, including XIAP, Bcl-xL and Bcl-2, and they were cooperative to upregulate the levels of BAD, a crucial proapoptotic member of BCL2 family (Figure 5C). Cytochrome C is released from mitochondria into cytoplasm when received pro-death signals and triggers ‘no-return’ mitochondria mediated apoptosis [35]. Combination of the two drugs obviously elevated the protein levels of Cytochrome C in comparison to the other three groups (Figure 5C). These data suggest that apatinib enhances the proapoptotic capability of chidamide in T-ALL cells probably through modulation of the protein levels of the BCL2 family members. 

### 3.5. Apatinib and Chidamide Are Active to Abrogate Leukemic Burden in a T-ALL PDX Model

To further confirm the antileukemia efficacy of the drug combination on T-ALL, a PDX model was generated via intravenous engraftment of primary T-ALL cells (2 × 10^6^ per mouse). Human CD45 expression in peripheral blood was monitored once every week and the treatment plan was initiated when its expression was ≥1%. These PDX mice were randomized into vehicle, apatinib (100 mg/kg/day), chidamide (10 mg/kg/day), and apatinib plus chidamide for consecutive 14-day treatment schedule. (Figure 6A). At the end of the experiment, all experimental mice were euthanized and their bone marrow (BM) and spleen (SP) specimens were collected to assess leukemia burden. Spleen weight and size were measured and photographed according to different treatment groups. Each single drug appeared to have no influence on spleen weight and size, whereas the coadministration of apatinib and chidamide was potent to ameliorate T-ALL disease-associated splenomegaly (Figure 6B,C). hCD45 positive and mCD45 negative cells were measured to assess the leukemia burden (Appendix A). Apatinib combined with chidamide remarkably lessened tumor burden in both BM (Figure 6D) and SP (Figure 6E), while apatinib and chidamide alone showed minimal impact on decreasing leukemia burden.

Immunohistochemistry and HE staining were applied to examine the expression levels of hCD45, a biomarker for leukemia infiltration, in liver tissues. Histological analysis showed that the liver tissues in the control group were infiltrated a large number of abnormal cells. However, apatinib or chidamide treatment alone decreased the leukemia infiltration in livers, and such effects was further enhanced in the combined treatment group (Figure 7A). Consistent with this finding, hCD45 staining was markedly decreased in the combination group compared to that in the single-drug groups or control group (Figure 7B).

TUNEL staining of liver tissues revealed a significant increase in apoptosis in the combined group (Figure 8A,B), in accordance with the in vitro findings. In addition, cotreatment of apatinib and chidamide strikingly blunted cell proliferative ability in T-ALL in vivo, evidenced by diminished expressions of PCNA and Ki-67, two cell proliferative indexes (Figure 8C–E). Altogether, combination therapy of apatinib and chidamide is effective in the management of T-ALL in vivo.

## 4. Discussion

T-ALL is a rare and aggressive lymphoid neoplasm with poor clinical outcome and limited therapeutic strategies, suggesting that the development of new treatment approaches is an unmet medical need for this disease entity. Our previous study has indicated that monotherapy with apatinib, a potent VEGFR2 antagonist, inhibits cell proliferation and promotes apoptosis in T-ALL in preclinical settings [16]. Data from clinical studies indicate that although apatinib alone shows excellent disease control, it results in moderate overall response rate (ORR) in diverse solid tumors [40], indicating that apatinib in combination with other drugs is indispensable to enhance its antitumor efficacy. In the present study, our results illustrated that apatinib enhanced the leukemia-killing effects of HDAC blockade with chidamide on T-ALL cellular models in vitro. What is more important, chidamide and apatinib were cooperative to alleviate leukemia burden in bone marrow and spleen of a T-ALL PDX model. This study provides a promising preclinical evidence of combining chidamide and apatinib as a potent alternative for the treatment of patients with the lethal T-ALL.

Apoptotic cell death can be triggered through cell surface death-receptors, the so-called death receptor or extrinsic pathway, or through the mitochondria, also known as intrinsic apoptosis. Mitochondrial apoptosis is highly regulated by members of the BCL-2 protein family. This family is grouped into antiapoptotic (e.g., XIAP, BCL2, BCLXL, etc.) and proapoptotic (e.g., BAD, BAK, BAX, etc.) subfamilies based on their roles in the intrinsic apoptosis pathway. During the apoptotic process, cytochrome C is released from mitochondrial intermembrane and favors cell death via activation and recruitment of pro-caspase-9 that in turn cleaves and activates the apoptotic executioner caspases-3 and -7 [39,41]. In agreement with its effects on acute myeloid leukemia (AML) and other solid tumor types, apatinib triggered the mitochondrial pathway of apoptosis in T-ALL cells partially by tilting the equilibrium of the BCL2 family proteins toward the proapoptotic members. Notably, such effect was significantly intensified when combining apatinib with HDAC inhibitor chidamide, as manifested by further decreasing BCL2, BCL-XL and XIAP protein levels, and increasing BAD expression. The imbalance of BCL2 family proteins induce by the drug combination subsequently resulted in enhancement of cytochrome C in T-ALL cells.

With constant detrimental stimulus, tumor cells activate self-defense mechanisms and thus adapt to such harmful environment. These mechanisms mainly include activation of DNA damage repair, downregulation of proapopototic factors and upregulation of prosurvival mediators. However, self-adaptation processes of tumor cells require continuous supply of ATP, a crucial and direct source of energy [42]. Metabolic reprogramming is a hallmark feature of neoplastic disorders, which not only provides sufficient energy by producing ATP but also offers essential biomasses for tumor cell survival and growth. Unlike other hematological malignancies, T-ALL is more reliant on the process of OXPHOS to produce ATP, and the amount of intracellular ATP levels determine the sensitivity of T-ALL cells to chemotherapeutic drugs [33]. We here demonstrated that the basal levels of mitochondrial aerobic respiration was minimally inhibited by both apatinib and chidamide treatment alone in T-ALL cells, as evidenced by slightly reduced OCR levels and ATP contents. This observation is consistent with previous reports that blocking VEGFR2 function sensitizes leukemia cells to chemotherapy through programming mitochondrial metabolism. It is also in keeping with the study that inhibiting HDACs with chidamide exerts antitumoral activity via suppression of mitochondrial aerobic metabolism in pancreatic cancer [43,44]. Such effect of decreasing OCR and ATP levels became more evident in the combination of apatinib and chidamide, indicating that the cooperative antileukemia effectiveness of the two drugs on T-ALL is potentially associated with perturbation of mitochondrial respiration process.

The aerobic respiration pathway in mitochondria mainly includes three pivotal steps, i.e., pyruvate metabolism, tricarboxylic acid cycle (TCA) and oxidative phosphorylation [36,45]. Each step contains a series of enzymatic reactions that require various enzymes. PDHA1, a key protein of pyruvate metabolism that controls the entry of pyruvate into mitochondria, was moderately downregulated with apatinib and chidamide treatment alone [46]. Of note, the two drugs acted additively to lessen the expression of PDHA1 in T-ALL cells. The expression level of MPC1, a carrier responsible for pyruvate transport [45], was significantly repressed by the apatinib and chidamide combination. Citric acid is the first product of mitochondrial aerobic oxidation and its enrichment was reduced by single drug treatment. Its production was cooperatively blocked with cotreatment of apatinib and chidamide. The TCA cycle is the final common oxidative pathway of carbohydrates, fats and amino acids. Multiple key enzymes, including CS, SDHD, MDH1 and COX10, have been implicated in this cycle and play important roles in ATP production. Abnormal expression of these enzymes is observed in multiple tumors, which is related to metabolic reprogramming in tumor metastasis [47,48]. Therefore, targeting them might be potential attractive therapeutic interventions. Indeed, compound 16c is a dual inhibitor of MDH1 and MDH2 [49] which has demonstrated significant in vivo antitumor efficacy and has shown effect on blocking mitochondrial respiration and hypoxia-induced HIF-1α accumulation. In accordance with the activity of apatinib and chidamide alone in the expression of PDHA1, we found that each drug decreased the abundance levels of CS, SDHD, MDH1 and COX10. Moreover, the combination of both drugs more powerful downregulated these key proteins in T-ALL cells. 

In this study, we provide clear evidence that apatinib and chidamide are additively cytotoxic against T-ALL in vitro and in vivo. Apatinib and chidamide are cooperative to activate the mitochondrial apoptosis pathway, repress mitochondrial respiration, diminish the enrichment of rate-limiting enzymes involved in the TCA and OXPHOS processes. Collectively, this preclinical work identifies apatinib combined with chidamide as a potential and promising therapeutic option for the treatment of T-ALL, while further clinical evaluations are required to further confirm their treatment interaction in T-ALL patients.

## Figures and Tables

**Figure 1 jpm-11-00977-f001:**
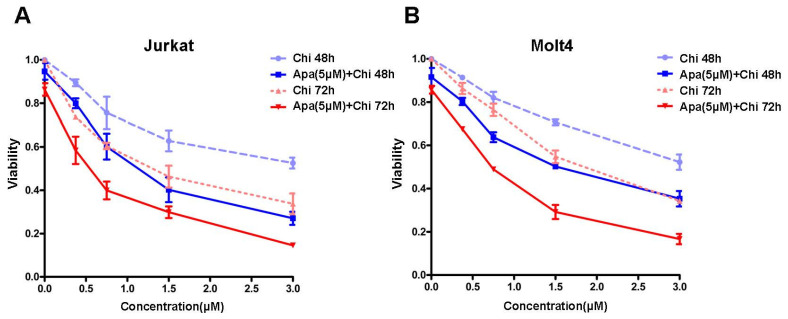
Apatinib potentiates the anti-leukemic activity of chidamide in T-ALL cell lines. T-ALL cell lines (Jurkat and Molt4) were separately treated with chidamide alone or in combination of 5 μM apatinib for both 48 h (**A**) and 72 h (**B**), and cell viability was then determined using a CCK8 assay. Data were presented as mean ± S.D. of triplicate assays.

**Figure 2 jpm-11-00977-f002:**
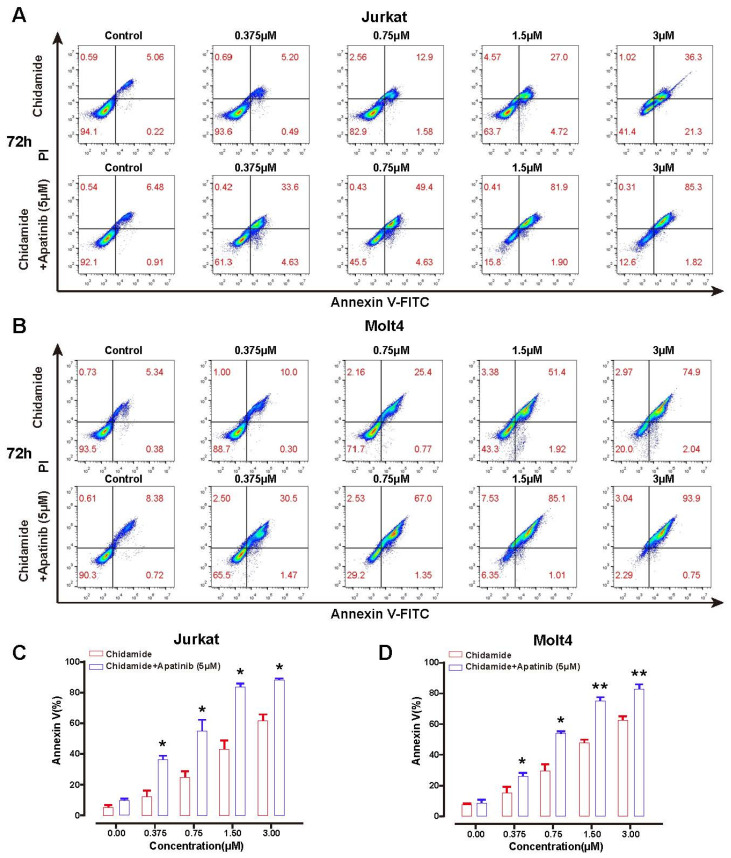
Apatinib and chidamide are cooperative to enhance cell apoptosis in T-ALL cells. After the exposure of Jurkat and Molt4 cells to chidamide with or without apatinib treatment for 72 h, the apoptotic cells were evaluated using Annexin V/PI staining by flow cytometry analysis. (**A**,**B**) were representative flow cytometry plots of apoptosis in Jurkat and Molt4 cells. Apoptosis analysis was used to analyze Jurkat (**C**) and Molt4 (**D**) cells. Values indicated mean ± S.D. for three independent experiments performed in triplicate. * *p* < 0.05, ** *p* < 0.01.

**Figure 3 jpm-11-00977-f003:**
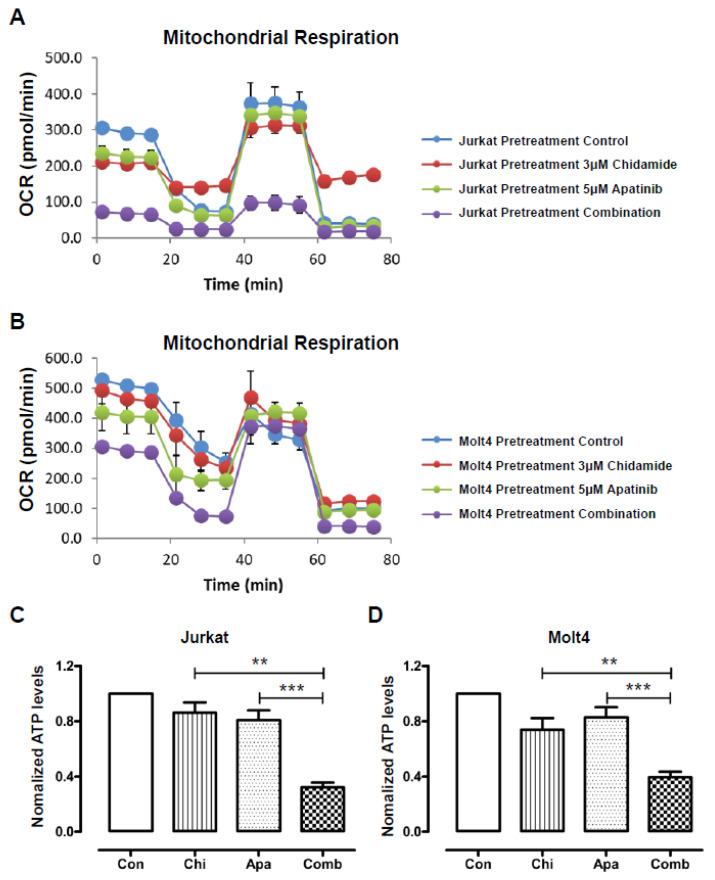
The cytotoxic effects of apatinib and chidamide on T-ALL cells are associated with a blockade of mitochondrial respiration. Molt4 and Jurkat cells were exposed to chidamide with or without apatinib treatment for 24 h. The pharmacological profile of OCR was monitored with a Seahorse XF96 analyzer in Jurkat (**A**) and Molt4 (**B**) cells. Metabolic inhibitors oligomycin, FCCP, rotenone and antimycin were added sequentially at different time points. Quantitative analysis of ATP generation in Jurkat (**C**) and Molt4 (**D**) cells. Data were shown as mean ± S.D. of three independent experiments in triplicate. ** *p* < 0.01, *** *p* < 0.001.

**Figure 4 jpm-11-00977-f004:**
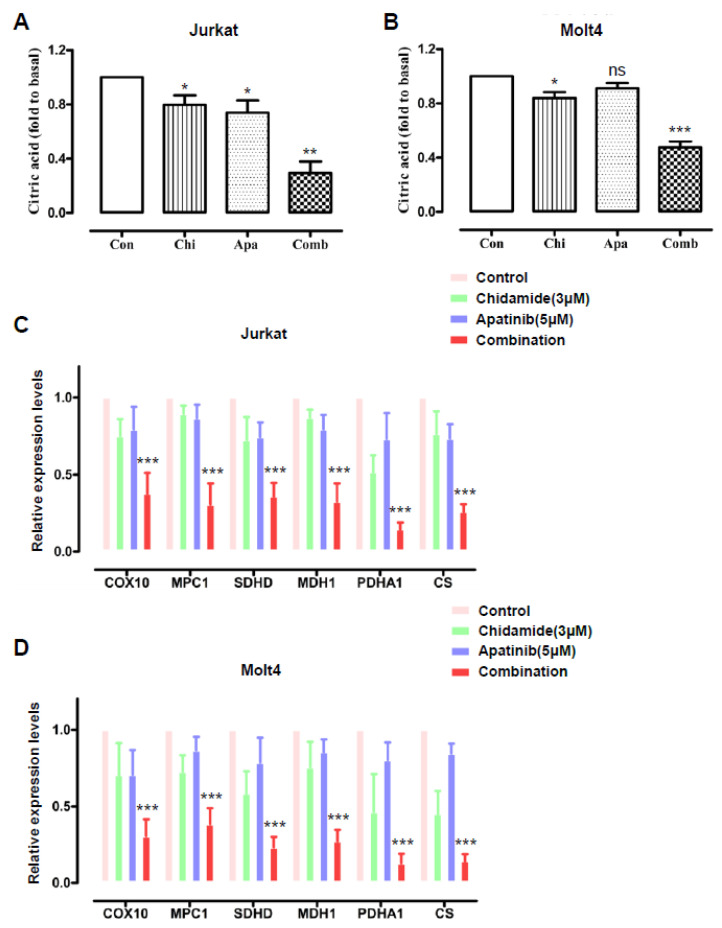
Cotreatment of apatinib and chidamide affects the citric acid cycle. Molt4 and Jurkat cells were exposed to chidamide (3 μM) and apatinib (5 μM) alone or in combination for 24 h. Citric acid (CA) production in Jurkat (**A**) and Molt4 (**B**) cells was examined. qRT-PCR analysis of the mRNA levels of PDHA1, MPC1, CS, SDHD, MDH1, and COX10 in Jurkat (**C**) and Molt4 (**D**) cells. Results were shown as mean ± S.D. of three independent experiments in triplicate. * *p* < 0.05, ** *p* < 0.01, *** *p* < 0.001, ns, not significant.

**Figure 5 jpm-11-00977-f005:**
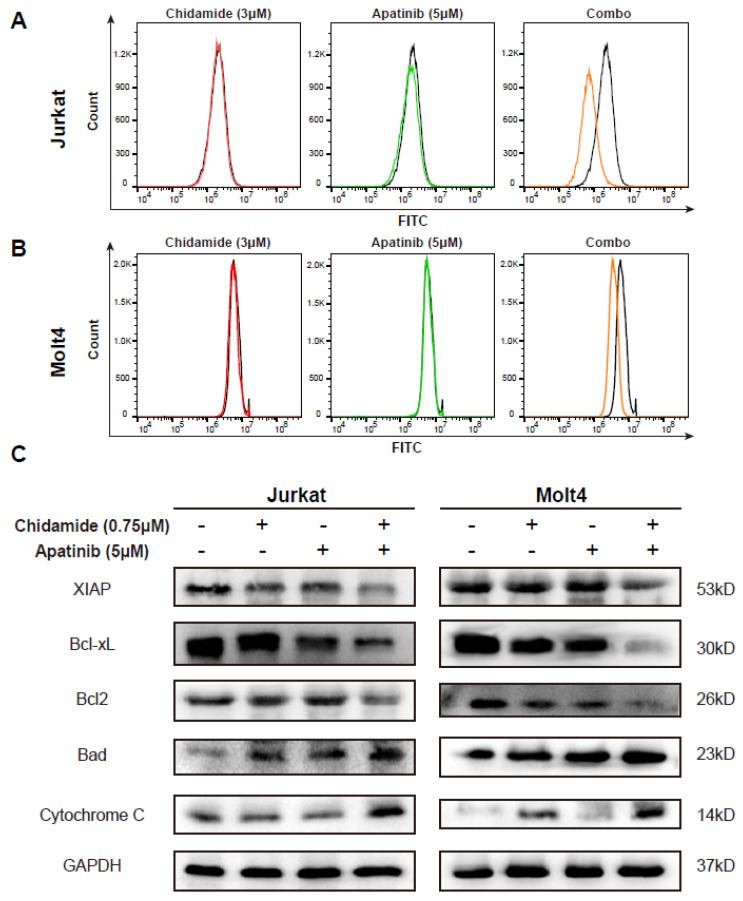
Apatinib potentiates chidamide induced apoptosis via deregulation of anti- and pro-apoptotic BCL-2 family components. Intracellular ROS levels was tested in Jurkat (**A**) and Molt4 (**B**) cells treated with chidamide in the presence or absence of apatinib for 24 h. (**C**) Western blot analysis of protein levels of BCL2, BCL-xL, XIAP, Bad and cytochrome C in both Molt4 and Jurkat cells exposed to chidamide (0.75 μM) and apatinib (5 μM) alone or in combination for 48 h. GAPDH was used as the loading control.

**Figure 6 jpm-11-00977-f006:**
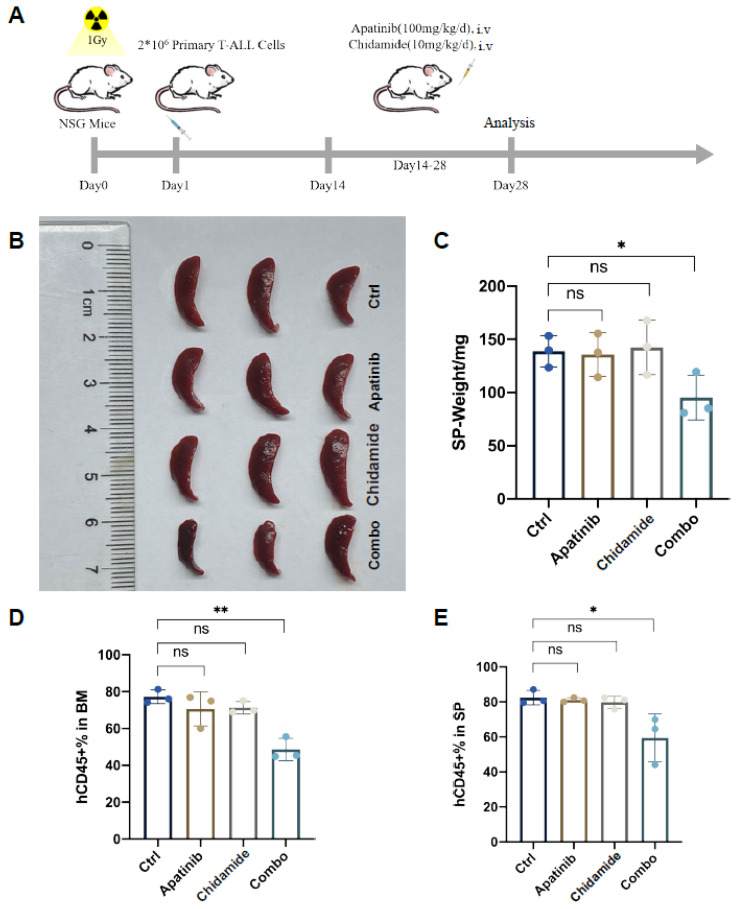
Apatinib and chidamide are active to abrogate leukemic burden in a T-ALL PDX model. (**A**) Overview of the in vivo treatment scheme. (**B**) Image of spleens harvested from PDX mice of distinct treatment groups. (**C**) Analysis of spleen weights of each group. Detection of human CD45 positive (hCD45+) and mouse CD45 negative cells in bone marrow ((**D**), BM) and spleen ((**E**), SP) using flow cytometry. * *p* < 0.05, ** *p* < 0.01, ns, not significant.

**Figure 7 jpm-11-00977-f007:**
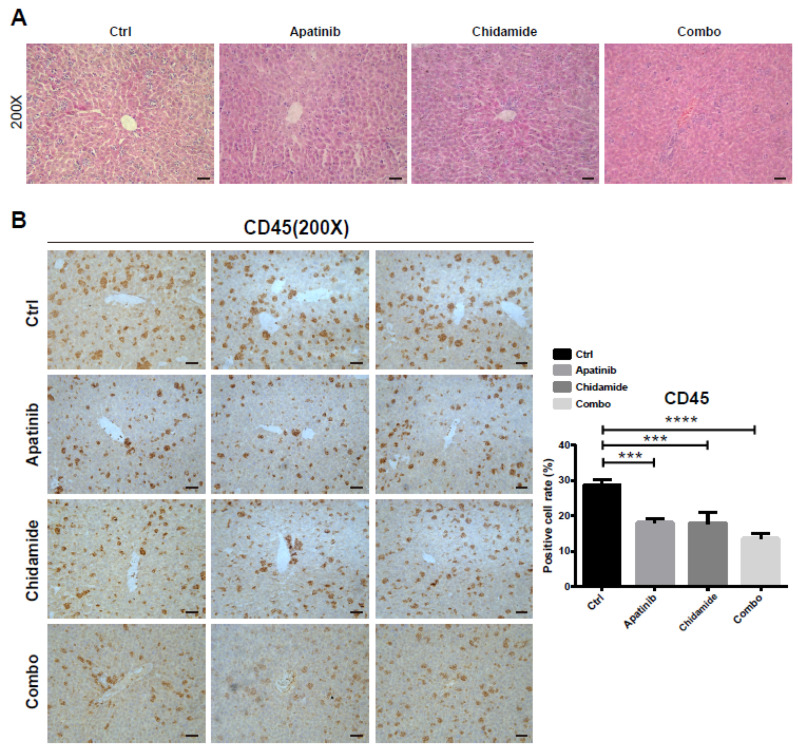
Analysis of leukemia infiltration in the liver tissues of PDX mice administrated with Apatinib and chidamide alone or combination for 2 weeks. (**A**) H&E staining analysis of leukemia burden in the liver sections of different treatment groups. (**B**) Representative immunohistochemistry micrographs showing hCD45 positive cell staining and quantification of hCD45 positive cell proportion in the liver sections obtained from different groups. *** *p* < 0.001, **** *p* < 0.0001.

**Figure 8 jpm-11-00977-f008:**
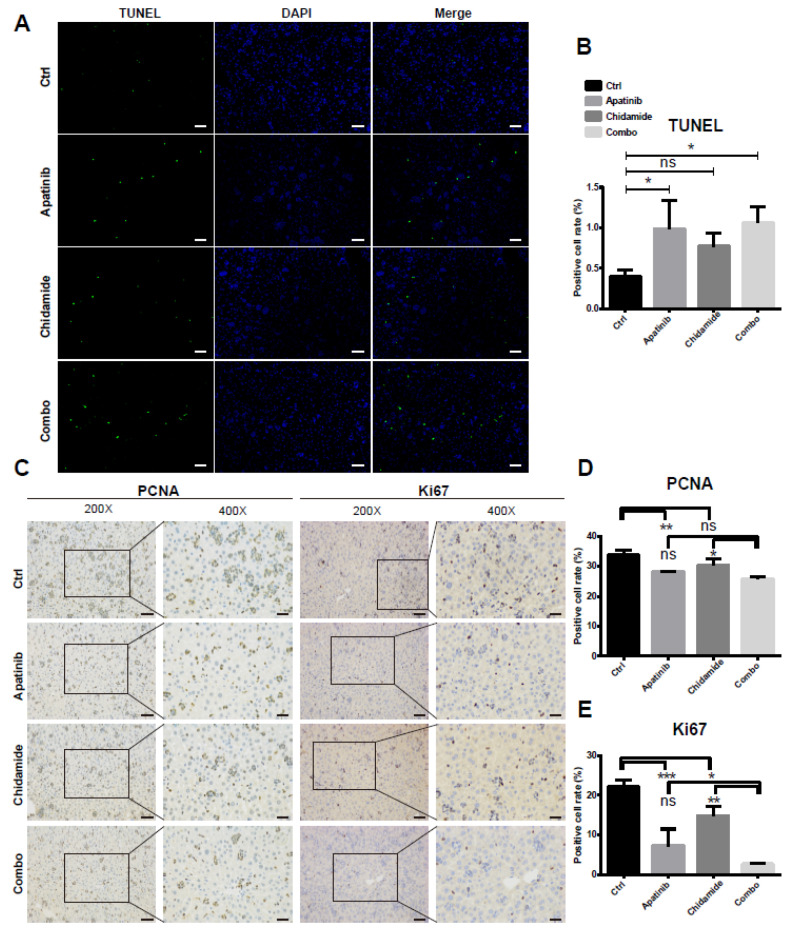
Antiproliferation effects of apatinib and chidamide on the PDX model. (**A**) Representative images of immunofluorescence staining of TUNEL in liver sections from PDX mice with different treatment. Images were taken with a Nikon microscope (original magnification: 200×). (**B**) Analysis of TUNEL staining in the liver tissues of PDX mice treated with vehicle, apatinib, chidamide and the combination of the two drugs. (**C**–**E**) Immunohistochemical staining for PCNA and Ki67 was performed to examine the antiproliferative activity of apatinib and chidamide in T-ALL in vivo. * *p* < 0.05, ** *p* < 0.01, *** *p* < 0.001, ns, not significant.

**Table 1 jpm-11-00977-t001:** The IC50 values of Jurkat and Molt4 cells treated with chidamide with or without Apatinib.

Cell Lines	IC_50_ (μM) of 48 h		Fold	*p* Value	IC_50_ (μM) of 72 h		Fold	*p* Value
	Single	Combination			Single	Combination		
Jurkat	3.22 ± 1.12	1.17 ± 0.28	2.75	0.037	1.34 ± 0.36	0.53 ± 0.22	2.52	0.031
Molt4	3.92 ± 1.28	1.52 ± 0.26	2.57	0.034	1.72 ± 0.08	0.73 ± 0.03	2.35	0.001

**Table 2 jpm-11-00977-t002:** Apoptosis percentage of Jurkat cells exposed to chidamide alone or in combination with apatinib for 48 and 72 h.

Treatment Timepoints	Chidamide (μM)	Apoptosis Rate (%)		*p* Value
		Chidamide	Apatinib (5 μM) + Chidamide	
48 h	0	4.63 ± 1.22	10.16 ± 4.17	0.1396
	0.375	9.34 ± 3.17	25.66 ± 8.97	0.0743
	0.75	16.03 ± 7.02	33.99 ± 9.23	0.0592
	1.5	21.00 ± 7.13	46.68 ± 7.91	0.0142
	3	30.50 ± 4.79	64.53 ± 5.47	0.0013
72 h	0	5.28 ± 1.93	9.75 ± 1.92	0.0566
	0.375	12.60 ± 6.88	36.66 ± 4.50	0.0104
	0.75	24.78 ± 7.27	55.52 ± 12.14	0.0282
	1.5	43.22 ± 10.52	84.34 ± 3.73	0.0132
	3	62.00 ± 6.80	88.24 ± 1.70	0.0169

**Table 3 jpm-11-00977-t003:** Apoptosis percentage of Molt4 cells exposed to chidamide alone or in combination with apatinib for 48 and 72 h.

Treatment Timepoints	Chidamide (μM)	Apoptosis Rate (%)		*p* Value
		Chidamide	Apatinib (5 μM) + Chidamide	
48 h	0	4.69 ± 0.84	7.33 ± 0.61	0.0544
	0.375	8.40 ± 2.05	17.92 ± 2.01	0.0345
	0.75	14.12 ± 2.33	28.50 ± 6.93	0.0372
	1.5	24.18 ± 3.33	45.02 ± 10.20	0.0397
	3	38.81 ± 2.60	59.34 ± 7.86	0.031
72 h	0	7.74 ± 0.96	8.85 ± 2.21	0.4913
	0.375	15.78 ± 6.09	25.86 ± 3.59	0.0435
	0.75	29.34 ± 5.89	53.95 ± 1.75	0.0129
	1.5	48.05 ± 2.93	74.82 ± 4.51	0.0019
	3	62.09 ± 3.82	82.33 ± 4.58	0.0046

## Data Availability

The data presented in this study are available in this published article and the associated Appendix A.

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
