# Peer review of "Therapeutic Interaction of Apatinib and Chidamide in T-Cell Acute Lymphoblastic Leukemia through Interference with Mitochondria Associated Biogenesis and Intrinsic Apoptosis"

_jpm, 2021, doi:10.3390/jpm11100977_

Round 1

Reviewer 1 Report

Progress in knowledge of cell mechanisms in neoplasms stimulated to use combined treatment based in those achievements. The authors examined the influence of two medicines Apatinib and  Chidamide in T-ALL cell lines Molt4 andJurkat as well on human T-ALL from patient xenografted on (PDX) mice. The results are valuable for future trials on patients. One thing that misses me is mention about the presence of VEGTR-2 receptor on cell lines and human T-ALL blasts. It would completed the manuscript.

Author Response

We really appreciate the reviewer’s encouraging comments and provide a constructive suggestion to impel us to improve our manuscript. In our prior published study, we have found that Jurkat cells expresses VEGFR2 and phosphorylated VEGFR2 (p-VEGFR-2) levels that are downregulated dose-dependently with Apatinib treatment alone (Deng M, et al. J Transl Med. 2018;16(1):47). However, the expression levels of VEGFR2 in Molt4 cells and human T-ALL blasts were not detected in that study. Due to the impact of Covid 19 outbreak in Xiamen, China, we are unable to perform experiments to investigate the expressions of VEGFR2 in Molt4 cells and human T-ALL blasts. In our future study, we will investigate the results of VEGFR2 expression in above mentioned cells.

Reviewer 2 Report

OVERVIEW

In this study, the authors aimed to assess the T-ALL therapeutic potential of combining two drugs, targeting distinct mechanisms: angiogenesis and histone deacetylation. The manuscript reports that the two drugs in combination were more effective in killing two T-ALL cell lines in vitro and in preventing PDX leukemogenesis than any of the drugs alone. Although several results are of interest, the paper suffers from several data presentation inconsistencies and errors. Moreover, as explained below, the results do not fully support the claim that Apatinib and Chidamide act synergistically to kill T-ALL cells.

MAJOR COMMENTS

  1. In the text, the authors use interchangeably the terms ‘cooperation’ and ‘synergism’, but the main message in the title, abstract and throughout the paper is that chidamide and apatinib act in synergy against T-ALL. Formally, ‘synergy’ means that the cooperation of two drugs leads to greater effects than the simple sum of the effects caused by each drug alone. The data in Figures 1 and 2 suggest that the effects of combining the two drugs are additive rather than synergistic. Even though the presented data support cooperation, to assess pharmacological synergy, experiments with both drugs applied over a range of concentrations are needed. Otherwise, the statement that apatinib and chidamide act in synergy remains an overstatement. Furthermore, the drugs apparently act upon independent cellular mechanisms, VEGFR2 and histone acetylation, so it becomes further elusive how blocking one of these mechanisms would potentiate the effects of blocking the other.
  2. In the abstract, it is mentioned that ‘numerous studies reported that Chidamide is highly effective against T-ALL in preclinical and clinical settings. However, there are only 3 reports in Pubmed, with one of them reporting non-statistically significant effects on a small number of patients (Guan et al, Leuk. Lymphoma, 2019). The effectiveness of chidamide against T-ALL is, therefore, far from settled. Furthermore, in Introduction, page 5, regarding chidamide anti-leukemia effects on T-ALL preclinical and clinical models, references 23 and 24 should be removed, because the first concerns AML and not T-ALL, and the second is about Romidepsin, a totally different drug. Also, it is written ’A series of studies emcompassing ours...’, but no references from these authors is provided, so this should be corrected.
  3. Its is stated that Chidamide or Apatinib monotherapy produced a lower level of citric acid in both Jurkat and Molt cells (page 16), but no statistical tests were applied in Figure 4 to support that statement.
  4. In Figure 5C, caspase 3 protein levels were shown, but much more relevant than full-length protein levels are the levels of cleaved caspase 3, which were not shown. A complete western blot picture should be provided to show  in the several experimental conditions the presence or absence of the low molecular weight caspase 3 fragment.
  5. Figure 5C indicated that chidamide was used at 0.75 μM concentration, but the legend indicates that 3 μM was used.
  6. The cellular analyses depicted in Figures 3, 4 and 5 (respiration, ATP levels, citric acid levels, gene expression, ROS levels, and apoptosis proteins) were performed with the highest concentrations of the two drugs (3 μM chidamide and 5 μM Apatinib) for 24 h. Although data at 24 h was not depicted, at 48 h and 72 h these concentrations induced strong cell death. This suggests that there was already a significant amount of cell death at 24 h. Without knowing if cells were viable at 24 h, it is difficult to conclude that the several cellular and molecular alterations were the cause of cell death, as concluded by the authors, or quite the opposite, the consequence. It is important to assess the viability levels of cells collected for molecular analyses, because apoptosis, or other forms of cell death, will trigger a number of cellular changes. If the analyses are done on a population of dying cells, it is difficult to dissect the sequence of events
  7. Regarding in vivo experiments, it is stated in Materials and Methods (page 10), and depicted in Figure 6A schematic, that the remaining 5 mice were utilized to determine the survival curve. However, no results were shown.
  8. The methodology to calculate the percentage of human CD45 cells in mouse organs (Figure 6D,E) was not provided. It is not clear if the percentage of cells was calculated in relation to total cells. The authors should show representative flow cytometry plots with the population gates to calculate percentage of cells in each mouse organ.
  9. Regarding histological analyses (or better immunohistochemical analyses), it is stated in Materials and Methods (page 11) that staining was scored based on both the percentage of positive tumor cells and staining intensity, but no analyses regarding staining intensity were shown. Moreover, it was not explained how the percentage of TUNEL, PCNA or Ki67 positive cells was calculated. Furthermore, the conclusions regarding these experiments should be cautious, because rather than reduced cell proliferation in the liver, the immunohistochemistry might be revealing reduce leukemic cell numbers in this organ. Liver infiltration may be reduced because of increased cell death in this organ or cell death in spleen and bone marrow. H&E histology and CD45 IHC should be performed to assess the percentage of human T-ALL cells in the liver of mice from each group.

MINOR COMMENTS

  1. What the black line represents in Figure 4A,B is not stated in legend.
  2. In Fig 6A, ‘i.g.’ is likely an error or is not explained what it means in the text.
  3. Panels in Figure 7A should be brighter to allow easier distinction of nuclei and TUNEL staining.
  4. The text contains an unusually high number of typos, e.g. promopting, emcompassing, affection, Cytochorme, comibnation, respcetively, synergitic.

Author Response

In this study, the authors aimed to assess the T-ALL therapeutic potential of combining two drugs, targeting distinct mechanisms: angiogenesis and histone deacetylation. The manuscript reports that the two drugs in combination were more effective in killing two T-ALL cell lines in vitro and in preventing PDX leukemogenesis than any of the drugs alone. Although several results are of interest, the paper suffers from several data presentation inconsistencies and errors. Moreover, as explained below, the results do not fully support the claim that Apatinib and Chidamide act synergistically to kill T-ALL cells.

Response: We appreciate the useful comments of the reviewer to help us improve our manuscript and to present our data in a better way. In response, we have carefully modified our manuscript and cautiously explained the results showed in the revised article.

MAJOR COMMENTS

  1. In the text, the authors use interchangeably the terms ‘cooperation’ and ‘synergism’, but the main message in the title, abstract and throughout the paper is that chidamide and apatinib act in synergy against T-ALL. Formally, ‘synergy’ means that the cooperation of two drugs leads to greater effects than the simple sum of the effects caused by each drug alone. The data in Figures 1 and 2 suggest that the effects of combining the two drugs are additive rather than synergistic. Even though the presented data support cooperation, to assess pharmacological synergy, experiments with both drugs applied over a range of concentrations are needed. Otherwise, the statement that apatinib and chidamide act in synergy remains an overstatement. Furthermore, the drugs apparently act upon independent cellular mechanisms, VEGFR2 and histone acetylation, so it becomes further elusive how blocking one of these mechanisms would potentiate the effects of blocking the other.

  Response: As pointed out by the reviewer, using ‘synergy’ to describe the therapeutic interaction of Chidamide and Apatinib might be not inappropriate. Therefore, the interaction of the two drugs has been amended as an additive effect on T-ALL cellular models.

Although Chidamide and Apatinib are designed to function as antitumor compounds mainly via blockade of histone deacetylase (HDAC) and VEGFR2 activities, they are potent to kill multiple tumor cells through different mechanisms of action. For example, Li Y, et al and Shi PC, et al reveal that dysregulation of DNA damage and repair contributes to the antileukemia effects of Chidamide alone or Chidamide-based regimen in acute myeloid and lymphoblastic leukemia (Li et al. Clinical Epigenetics (2017) 9:83; Shi P, et al. Pharmacogenomics. 2017; 18(13):1259-1270). In terms of Apatinib, this drug also could exert its antitumor activity through perturbation of diverse molecular basis except inhibition of VEGFR2 expression (Feng H, et al. Cell Death Dis. 2018;9(10):1030; Zhao S, et al. Cancer Immunol Res. 2019 Apr;7(4):630-643; etc.). Therefore, together with our investigations showed in the manuscript files, we believe that the potential mechanism for the therapeutic additive effects of Chidamide and Apatinib on T-ALL cells can be achieved through influencing mitochondria associated biogenesis and intrinsic apoptosis, rather than blocking the activity of HDAC and VEGFR2.

  1. In the abstract, it is mentioned that ‘numerous studies reported that Chidamide is highly effective against T-ALL in preclinical and clinical settings. However, there are only 3 reports in Pubmed, with one of them reporting non-statistically significant effects on a small number of patients (Guan et al, Leuk. Lymphoma, 2019). The effectiveness of chidamide against T-ALL is, therefore, far from settled. Furthermore, in Introduction, page 5, regarding chidamide anti-leukemia effects on T-ALL preclinical and clinical models, references 23 and 24 should be removed, because the first concerns AML and not T-ALL, and the second is about Romidepsin, a totally different drug. Also, it is written ’A series of studies emcompassing ours...’, but no references from these authors is provided, so this should be corrected.

Response: The comments are valid and helpful. In response, we have amended these inappropriate words or sentences and marked them in red throughout the entire main text. The improperly cited references have been removed or replaced with proper ones (Ref.14,25) shown in the Reference section of the revised manuscript. In addition, we have added several appropriate cites (Ref. 30,31) into the reviewer mentioned place in the introduction section.

  1. It is stated that Chidamide or Apatinib monotherapy produced a lower level of citric acid in both Jurkat and Molt cells (page 16), but no statistical tests were applied in Figure 4 to support that statement.

Response: Thanks for the kind reminder. As requested, we have compared each single drug with the control group and calculated the statistical significance. In Molt4 cells, Chidamide alone led to significant reduction of citric acid (CA) levels with P < 0.05. However, Apatinib treated Molt4 cells showed slight CA decrease with no statistical difference as compared with the control group. Regarding Jurkat cells, both Chidamide and Apatinib treatment alone significantly lessened the CA levels with P < 0.05. Accordingly, the results of these statistical tests have been added into the revised Figure 4A-B.

  1. In Figure 5C, caspase 3 protein levels were shown, but much more relevant than full-length protein levels are the levels of cleaved caspase 3, which were not shown. A complete western blot picture should be provided to show in the several experimental conditions the presence or absence of the low molecular weight caspase 3 fragment.

Response: The reviewer is correct that the cleaved form of caspase 3 might have more predictable implications for apoptosis than full-length caspase 3 protein. However, reduction of full-length caspase 3 is accompanied by increased caspase 3 in cleaved form, thus it is considered an indirect evidence of cell apoptosis. In previous reports, either Apatinib (Pan Q, et al. J BUON. 2019;24(1):374-381) or Chidamide (Zhang H, et al. Mol Med Rep. 2021;23(5): 308) alone could cut full-length caspase 3 into cleaved protein, indicating that the effect of the two drug combination on full-length caspase 3 might be translated into elevation of cleaved caspase 3. The limited revision duration and current COVID-19 outbreak in Xiamen prevent us from performing new experiments which will include low molecular weight caspase 3 fragment with full-length proteins. Hope the explanation will at least partially meet the reviewer’s expectation.

  1. Figure 5C indicated that chidamide was used at 0.75 μM concentration, but the legend indicates that 3 μM was used.

Response: Thank you for pointing out the inconsistency of Chidamide concentration between the Figure 5C and the corresponding Figure legend. After carefully checking the orignal data, we confirmed that the Chidamide dose used in Figure 5C was 0.75μM and the relevant Figure legend have been modified in red in page 34, line 742.

  1. The cellular analyses depicted in Figures 3, 4 and 5 (respiration, ATP levels, citric acid levels, gene expression, ROS levels, and apoptosis proteins) were performed with the highest concentrations of the two drugs (3 μM chidamide and 5 μM Apatinib) for 24 h. Although data at 24 h was not depicted, at 48 h and 72 h these concentrations induced strong cell death. This suggests that there was already a significant amount of cell death at 24 h. Without knowing if cells were viable at 24 h, it is difficult to conclude that the several cellular and molecular alterations were the cause of cell death, as concluded by the authors, or quite the opposite, the consequence. It is important to assess the viability levels of cells collected for molecular analyses,

Response: The reviewer is correct. Based on our previous studies (Deng M, et al. J Transl Med. 2018;16(1):47; Shi P, et al. Pharmacogenomics. 2017;18(13):1259- 1270.), apatinib and chidamide alone showed a time-dependent cytotoxic effects against T-ALL cells. This observation indicates that treatment T-ALL cells with the two drugs individually might be less effective at 24h exposure than 48 or 72h exposures. Our preliminary data shown below verified that neither chidamide or chidamide combined with apatinib resulted in significant loss of cell viability in both Jurkat and Molt4 cells. Therefore, we believe the selected concentration of apatinib and chidamide alone or in combination might not lead to evident cell death in both T-ALL lines at 24h treatment and then affect the results depicted in Figures 3, 4 and 5.

  1. Regarding in vivo experiments, it is stated in Materials and Methods (page 10), and depicted in Figure 6A schematic, that the remaining 5 mice were utilized to determine the survival curve. However, no results were shown.

Response: We are really appreciate the reviewer’s carefulness and rigorousness with respect to our manuscript and related files. At the beginning of this study, we planned to evaluate the survival impact of the combination of Apaitnib and Chidamide on T-ALL PDX mouse models, while limited financial budgets and Covid-19 outbreak prevented us from continuing the survival curve assay. In our future studies, we will investigate whether the anti-leukemia effects of this drug combination in T-ALL PDX models would translate into survival benefits in T-ALL preclinical and clinical settings.

  1. The methodology to calculate the percentage of human CD45 cells in mouse organs (Figure 6D, E) was not provided. It is not clear if the percentage of cells was calculated in relation to total cells. The authors should show representative flow cytometry plots with the population gates to calculate percentage of cells in each mouse organ.

Response: These comments are valid and valuable to improve our manuscript. As required, the methodology of how to gate and count the hCD45 positive percentage of the PDX model has been provided in the section of Materials and Methods in the revised version (Page 11, lines 232-235 and lines 239-245). In this assay, cells collected from bone marrow (BM) and spleen (SP) of the PDX mice were stained with Biotin anti-human CD45 antibody and BV421 Anti-Mouse CD45 antibody for 30 min and then referred to flow cytometry analysis. The leukemia burden in BM and SP was defined as human CD45-positive and mouse CD45-negative cells which were gated from almost total cells. The representative flow cytometry plots in the experiment has been placed below.

  1. Regarding histological analyses (or better immunohistochemical analyses), it is stated in Materials and Methods (page 11) that staining was scored based on both the percentage of positive tumor cells and staining intensity, but no analyses regarding staining intensity were shown. Moreover, it was not explained how the percentage of TUNEL, PCNA or Ki67 positive cells was calculated. Furthermore, the conclusions regarding these experiments should be cautious, because rather than reduced cell proliferation in the liver, the immunohistochemistry might be revealing reduce leukemic cell numbers in this organ. Liver infiltration may be reduced because of increased cell death in this organ or cell death in spleen and bone marrow. H&E histology and CD45 IHC should be performed to assess the percentage of human T-ALL cells in the liver of mice from each group.

Response: We apologize for the confused description with regard to the staining score for the immunohistochemistry (IHC) analysis. In the revised version, we have carefully modified them in page 12, lines 254-256 and lines 259-263. In brief, the Eclipse Ci-L microscope (Nikon, Japan) was used to select the target area of the liver slices for 200x imaging, and the entire field of the slices was captured to ensure that the background light of each image is consistent. After obtaining the imaging, Image-Pro Plus 6.0 analysis software was employed to measure the number of positive cells in randomized three fields of view in each slice and to calculate the corresponding total cell number. Finally, the following formula was utilized to calculate the positive rate (%).

Positive rate (%) = number of positive cells/ number of total cells*100.

Leukemia burden in the liver tissues of PDX mice was analyzed with both HE and CD45 staining assays. In line with the results of hCD45 positive cells in BM and SP,  the leukemia burden in liver was obviously alleviated by the combination of apatinib and chidamide. Accordingly, the corresponding result descriptions have been added into the main text seen in page 18, lines 379-386, and the relevant Figures have been supplemented in the Supplementary Figure 2

MINOR COMMENTS

1. What the black line represents in Figure 4A,B is not stated in legend.

Response: In fact, we found there were 2 black lines showed in Figure 4A,B. One represents X-axis and the other one represents Y-axis,

2. In Fig 6A, ‘i.g.’ is likely an error or is not explained what it means in the text.

 Response: This comment is totally right. We feel sorry for this typos showed in the Figure 6A. Indeed, the ‘i.g.’ should be replaced with ‘i.v.’ which means the T-ALL PDX model mice were intravenously administered with Apaitnib and Chidamide alone or in combination. Accordingly, we have changed ‘i.g.’ into ‘i.v.’ in the revised Figure 6A.

3. Panels in Figure 7A should be brighter to allow easier distinction of nuclei and TUNEL staining.

Response: As requested, the orignial Figure 7A has been replaced with higher resolution image figures, in which nuclei and TUNEL staining are much easier to recognize.

4. The text contains an unusually high number of typos, e.g. promopting, emcompassing, affection, Cytochorme, comibnation, respcetively, synergitic.

Response: Thank you for pointing out these spelling drawbacks showed in the manuscript. In the revised one, we have carefully double-checked such shortcomings and referred the manuscript to a well-known English-editing service company to help us correct the grammar and typo errors. Hope these modifications will meet the requirements of the JPM.

Round 2

Reviewer 2 Report

OVERVIEW

The authors addressed most of remarks made and have improved significantly the quality of the manuscript. I have still a few remarks before final acceptance.

COMMENTS

  1. (Related to previous comment 4) The CA9662S antibody from CST is supposed to detect both full-length (35 kDa) and cleaved (17-19 kDa) Caspase 3 protein. If the authors did not detect cleaved caspase 3 fragment with this antibody, then the cells were not undergoing apoptosis or it was not working properly. If the authors cannot guarantee the antibody is working well in apoptotic cells, then the caspase 3 data should be removed and the text changed accordingly.
  2. (Related to previous comment 6) The cell viability at 24 h is important for interpretation of other experimental results, so the graphs the authors provided should be included in the manuscript as supplementary figures. These graphs may explain the caspase 3 results. At 24 h, and 0.75 uM, most cells (>80%) seem to be viable. Viable cells should have no caspase 3 cleavage because this is a late apoptotic event.
  3. (Related to previous comment 8) The flow cytometry plots are of interest for readers, so they should be included in the manuscript as supplementary figures.
  4. (Related to previous comment 9) I appreciated the H&E and CD45 IHC data provided as supplementary data. I recommend the CD45 IHC images and graph to be shown as main figure in the article. I think it will enrich the paper.

Author Response

Reviewer 2:

The authors addressed most of remarks made and have improved significantly the quality of the manuscript. I have still a few remarks before final acceptance.

Response: We appreciate the positive comments of the reviewer that we have well addressed the main concerns and markedly polished our manuscript in the revised revision. The following point-to-point responses may further address the minor concerns of the reviewer about this paper.

COMMENTS

1.(Related to previous comment 4) The CA9662S antibody from CST is supposed to detect both full-length (35 kDa) and cleaved (17-19 kDa) Caspase 3 protein. If the authors did not detect cleaved caspase 3 fragment with this antibody, then the cells were not undergoing apoptosis or it was not working properly. If the authors cannot guarantee the antibody is working well in apoptotic cells, then the caspase 3 data should be removed and the text changed accordingly.

Response: The reviewer is correct. Currently, we cannot make sure whether the anti-caspase-3 antibody works well or not, and we have been kept from performing further experiments to probe the cleaved form of caspase-3 using an anti-cleaved-caspase-3 monoclonal antibody because of the unexpected outbreak of COVID-19 in Xiamen. Therefore, the full-length caspase-3 has been deleted from the original Figure 5C. Accordingly, we have modified the text and figure legend statement about the expression levels of full-length caspase-3.

  1. (Related to previous comment 6) The cell viability at 24 h is important for interpretation of other experimental results, so the graphs the authors provided should be included in the manuscript as supplementary figures. These graphs may explain the caspase 3 results. At 24 h, and 0.75 uM, most cells (>80%) seem to be viable. Viable cells should have no caspase 3 cleavage because this is a late apoptotic event.

Response: These are valid comments. We appreciate the precise understanding of the reviewer with regard to the association between the viability and apoptosis of T-ALL cells treated with Chidamide with or without Apaitnib for 24h. On the basis of the 24h viability of both T-ALL cell lines, we believe that exposure T-ALL cells to Chidamide alone or in combination with Apaitnib for 24h could have minimal effect on cell apoptosis. Therefore, the cell viability data of T-ALL cell lines obtained from 24h treatment have been introduced into the new supplementary Figure 1 as requested.

  1. (Related to previous comment 8) The flow cytometry plots are of interest for readers, so they should be included in the manuscript as supplementary figures.

Response: Thank you for the encouraging comment. In response, the flow cytometry plots showing the percentage of hCD45 positive and mCD45 negative cells in the BM and SP have been transferred to the new supplementary Figure 3.

  1. (Related to previous comment 9) I appreciated the H&E and CD45 IHC data provided as supplementary data. I recommend the CD45 IHC images and graph to be shown as main figure in the article. I think it will enrich the paper.

Response: We are thankful for the constructive suggestion of the reviewer. As suggested, the figures of H&E and CD45 IHC have been moved from the original supplementary Figure 3 to the new Figure 7 in the revised files. The text descriptions also have been accordingly modified and marked in red seen in page 18, lines 382 and 384. In addition, the following figures are renumbered in order.
